# USP28: Oncogene or Tumor Suppressor? A Unifying Paradigm for Squamous Cell Carcinoma

**DOI:** 10.3390/cells10102652

**Published:** 2021-10-04

**Authors:** Cristian Prieto-Garcia, Ines Tomašković, Varun Jayeshkumar Shah, Ivan Dikic, Markus Diefenbacher

**Affiliations:** 1Protein Stability and Cancer Group, Department of Biochemistry and Molecular Biology, University of Würzburg, 97074 Würzburg, Germany; 2Comprehensive Cancer Centre Mainfranken, 97074 Würzburg, Germany; 3Molecular Signaling Group, Institute of Biochemistry II, Goethe University Frankfurt, 60590 Frankfurt am Main, Germany; tomaskovic@med.uni-frankfurt.de (I.T.); shah@med.uni-frankfurt.de (V.J.S.); dikic@biochem2.uni-frankfurt.de (I.D.); 4Buchmann Institute for Molecular Life Sciences, Goethe University Frankfurt, 60590 Frankfurt am Main, Germany; 5Mildred Scheel Early Career Center, 97074 Würzburg, Germany

**Keywords:** USP28, SCC, USP25, FBXW7, Tp63, c-MYC, ΔNp63, p53, cancer, DUB inhibitor, squamous

## Abstract

Squamous cell carcinomas are therapeutically challenging tumor entities. Low response rates to radiotherapy and chemotherapy are commonly observed in squamous patients and, accordingly, the mortality rate is relatively high compared to other tumor entities. Recently, targeting USP28 has been emerged as a potential alternative to improve the therapeutic response and clinical outcomes of squamous patients. USP28 is a catalytically active deubiquitinase that governs a plethora of biological processes, including cellular proliferation, DNA damage repair, apoptosis and oncogenesis. In squamous cell carcinoma, USP28 is strongly expressed and stabilizes the essential squamous transcription factor ΔNp63, together with important oncogenic factors, such as NOTCH1, c-MYC and c-JUN. It is presumed that USP28 is an oncoprotein; however, recent data suggest that the deubiquitinase also has an antineoplastic effect regulating important tumor suppressor proteins, such as p53 and CHK2. In this review, we discuss: (1) The emerging role of USP28 in cancer. (2) The complexity and mutational landscape of squamous tumors. (3) The genetic alterations and cellular pathways that determine the function of USP28 in squamous cancer. (4) The development and current state of novel USP28 inhibitors.

## 1. Deubiquitinase USP28

Ubiquitination is one of the most relevant posttranslational modifications and regulates critical cellular processes, including protein degradation, cell cycle progression, autophagy, transcription or DNA repair. Ubiquitin contains seven different lysine residues (K6, K11, K27, K29, K33, K48, and K63) and N-terminal methionine (M1) groups that serve as secondary ubiquitination sites to induce different types of ubiquitin chains on substrates [1]. Protein ubiquitination is a reversible process, occurring through the action of deubiquitinating enzymes (DUBs) that cleave the bond between ubiquitin and the target proteins [2]. Notably, some DUBs display a certain degree of selectivity towards specific types of ubiquitin chains, while others show a broad promiscuity [3].

Ubiquitin-specific peptidase 28 (USP28) belongs to the largest DUB family (USPs) and has been reported to regulate the ubiquitination status of several targets involved in proliferation, DNA repair, apoptosis and oncogenesis. In contrast to other USP enzymes, USP28 preferentially deubiquitinates K11, K48, and K63 ubiquitin chains [4]. The human USP28 gene maps to chromosome 11q23 and it encodes a protein with 1077 amino acids. USP28 can generate different isoforms by alternative splicing. The canonical USP28 sequence, also called the shorter isoform, does not contain the 62 amino acids of exon 19a (Figure 1A) and it is ubiquitously expressed in all tissues. Alternatively, the tissue-specific isoform containing exon 19a was only expressed in the muscle, heart and brain [5].

USP28 was identified through a homology search for USP25, as they share a 51% sequence identity [6]. Similar to other DUB ohnolog pairs, USP28 and USP25 were generated from ancient whole-genome duplication, and they have shown limited divergence upon the duplication event [7]. The deubiquitinases USP25 and USP28 have similar topological structures, comprising USP domains, ubiquitin-associated domains (UBAs), sumo interacting domains (SIMs) and two ubiquitin-interacting motifs (UIMs) at the N-terminal region [3] (Figure 1A,B). However, USP25 and USP28 are not functionally redundant, and the protein localization of both the DUBs differs. USP25 is located in the cytosol, whereas USP28 is a nuclear protein. Additionally, USP28 can form functionally active dimers, but only USP25 can form inactive tetramers with two dimers [8,9].

The expression and enzymatic activity of USP28 is strongly regulated by several cellular processes (Table 1) in a context-specific manner. In particular, posttranslational modifications tightly regulate USP28 activity (Figure 1A). Upon DNA damage, the phosphorylation of USP28 on serine 67 and serine 714 by the kinase ATM increases its enzymatic activity [10]. Apart from phosphorylation, Sumoylation can also regulate USP28 enzymatic activity. Sumoylation is defined as the reversible conjugation of small-ubiquitin-related modifier molecules (SUMOs) on a substrate protein [11,12]. N-terminal sumoylation of USP28 decreases its enzymatic activity [4,13]. SENP1 strongly desumoylates USP28 upon hypoxia [14]. As reported for other DUBs [15,16], one cannot exclude the possibility that USP28 deubiquitinates itself, thereby avoiding proteasomal degradation and, in consequence, regulating its own stability. Supporting this hypothesis, recent publications observed reduced USP28 protein levels upon pharmacological inhibition [17].

Independent of posttranslational modifications, other mechanisms have been reported that can regulate the activity or expression of USP28 in cells (Table 1). As an example, the cleavage of USP28 by Caspase-8 inactivates the DUB and is required to overcome the p53-dependent G2/M DNA damage checkpoint [18]. Alternatively, histone deacetylase 5 (HDAC5) promotes USP28 stability and positively regulates the protein abundance of the Lysine-specific histone demethylase 1A (LSD1) [19]. The expression of USP28 can also be transcriptionally regulated, as the oncogenic transcription factors c-JUN and c-MYC bind to the USP28 promotor and positively regulatie its expression [20,21]. Lastly, it was reported that different microRnas (miR) and circular RNAs (circRNAs) regulate the activity of USP28. The microRnas miR-92b-3p, miR-216b, miR-500a-5p, and miR-3940-5p negatively regulate USP28 expression blocking its translation [22,23,24,25] while the circRNA FBXW7 reduce USP28 activity, decreasing substrate recognition [26].

USP28 was identified as the first DUB capable of antagonizing FBXW7, thus regulating c-MYC stability (Figure 2A) [27]. FBXW7 is a member of the F-box family of E3-ligases. FBXW7 is part of the Skp1-Cdc53/Cullin-F-box (SCF) protein complex and mediates the binding and potential ubiquitination of substrates. FBXW7 contains seven tandem WD40 repeats, which can form a β-propeller structure, allowing the binding of FBXW7 to the cysteine protease domain (CPD) region of a phosphorylated target protein. FBXW7 is an essential E3-ubiquitin ligase that regulates the stability of important oncogenic transcription factors, such as c-MYC, c-JUN, NOTCH1, and ΔNp63 [28,29].

FBXW7 is frequently mutated or deleted in malignant human tumors [30]. Missense mutations within the WD40 domain of FBXW7 or in the CPD domain of a substrate disrupt the FBXW7-mediated ubiquitination of the target proteins [31,32]. Using genetically engineered mouse models (GEMMs), previous studies have demonstrated that the loss of FBXW7 significantly accelerated the progression of murine colorectal tumors by accumulation of oncoproteins c-JUN, c-MYC, CCNE1, and NOTCH1. Alternatively, the deletion of USP28 counteracted the loss of FBXW7, thereby decreasing the oncoprotein stability of the ligase substrates and increasing the life expectancy of the animals from 70 days to 122 days. Accordingly, FBXW7 mutant tumors require USP28 activity to support and maintain the malignant transformation during colorectal oncogenesis [21,33]. This example highlighted the fact that oncogenic alterations in the UPS system require remodeling of the ubiquitin system, creating new oncogenic dependencies that can be therapeutically targeted.

USP28 not only stabilizes the substrates of FBXW7 (Figure 2A), but also deubiquitinates FBXW7 itself upon autocatalytic ubiquitination (Figure 2B) [34]. This dual regulation of USP28 and FBXW7 allows for control of the protein abundance of common substrates, such as c-MYC and NOTCH1. In addition to the previously mentioned substrates, USP28 stabilizes other targets involved in carcinogenesis [35], such as HIF-1α, LSD1, STAT3, CD44, FOXC1, LIN28A and CCNE (Table 2). Furthermore, recent studies have demonstrated that USP28 deubiquitinates and stabilizes important tumor suppressors, such as p53 and CHK2 [18,36].

USP28 was originally identified as a protein involved in DNA damage response (DDR), interacting and stabilizing the double-strand break repair protein 53BP1 [10]. This interaction results in the phosphorylation of USP28 on serine 67 and 714 upon exposure to ionizing radiation in an ATM-dependent manner [10,37]. Furthermore, USP28 also regulates G2/M DNA damage checkpoint by preventing CLASPIN degradation upon ubiquitination of the E3-ligase anaphase-promoting complex/cyclosome (APC/C) [38]. USP28 is involved in the stabilization of several other proteins related to DDR pathways, namely MDC1 and CHK2 [10,36].

For Squamous Cell Carcinoma (SCC), USP28 function was recently clarified [17]. In SCC, USP28 is strongly expressed and stabilizes the essential squamous transcription factor ΔNp63, together with other important oncoproteins, such as NOTCH1, c-MYC and c-JUN [17]. USP28 gene expression correlated with poor prognosis and shortened lifespans in SCC patients [17]. Notably, SCC requires high levels of the axis USP28-ΔNp63 to maintain the malignant phenotype and its pharmacologic inhibition dramatically reduces the number of SCC tumors in lung cancer mouse models [17].

## 2. Squamous Cell Carcinoma

SCC is one of the most complex and heterogeneous entities. While driver mutations and genetic alterations can vary extensively, all SCC have share an epithelial origin and squamous histology [53,54]. The epithelium is a basic tissue composed of three different types of cells classified by shape and function:○Cuboidal cells: These are cube-shaped cells with large, spherical central nuclei (Figure 3A). Cuboidal cells provide protection and mechanical support. Notably, they can differentiate and form secretory glands. Kidney or salivary glands are recovered by cuboidal cells upon tissue damage.○Squamous cells: The squamous epithelium is a selective permeable layer formed by a delicate line of thin and flat cells (Figure 3B). The esophagus or oral cavity is covered by squamous epithelial cells.○Columnar cells: These cells are taller than squamous or cuboidal cells and present an oval nucleus (Figure 3C). Columnar cells facilitate movement across the epithelial barrier, and some tissues with the columnar epithelium have cells with cilia. Intestine or uterus is covered by columnar cells.

The epithelium is considered simple if it contains one layer of cells (Figure 2B,C and Figure 3A) and stratified when it contains two or more layers of cells (Figure 3D). The higher the number of layers, the more protective the epithelial tissue. One clear example of stratified epithelial tissue is the skin, where the squamous epithelial tissue of the skin is arranged in many layers to act as a barrier, offering strong mechanical protection. Tumors arising from epithelial tissues are called carcinomas [55]. Stratified epitheliums can comprise different types of epithelial cells assembling complex tissues [56]. The epithelial carcinoma is considered the most common type of cancer, and it can be subdivided as follows:○Squamous cell carcinoma (SCC): This is also called epidermoid carcinoma and it is composed of squamous cells. As mentioned earlier, squamous cells are flat cells that can be present in many different organs of the body. Accordingly, squamous tumors can be developed in several different tissues such as in the lung, skin, esophagus, prostate, pancreas, cervix, vulva, thyroid, head and neck, penis, and bladder. The precise “cell of origin” for SCC has not yet been identified, and various cellular pools have been suggested for the same [53,57].○Adenocarcinoma (ADC): ADC tumors arise from epithelial cells with secretory properties or glandular cells. ADC can be formed in different organs such as the lungs, breasts, esophagus, colon, stomach, and prostate [58]. Carcinomas can be considered adenosquamous (ADSCC) when the tumor is presenting the expression of markers and histopathological features of SCC and ADC. Each type of cell must constitute at least 10% of the tumor bulk to be diagnosed as ADSCC [59].○Transitional cell carcinomas: These carcinomas arise from the transitional epithelium, which is a type of stratified epithelium tissue composed of multiple layers of epithelial cells with different morphologies. Transitional carcinomas can be contracted or expanded to adapt themselves to the degree of distension needed by the tissue. Hence, this tumor type usually occurs in organs that form part of the urinary system, such as the bladder [60].○Basal cell carcinomas: These tumors originate in the cells of the basal layer, located at the lower part of the epidermis. Basal cell carcinoma is the most common type of skin cancer [61].

SCCs comprise one of the largest groups of cancer types. They are histologically characterized by the presence of intercellular bridges, keratinization, and squamous pearls composed of keratins. ∆NP63 is the main histopathological marker used for classifying SCC [62,63]. The common relevant secondary SCC markers are Keratin 5/6 (KRT5/6), Podoplanin, and Keratin 14 (KRT14) [64,65].

Lung cancer, one of the primary causes of cancer-related deaths in the world, is a suitable model to study the differential epithelial-derived tumor types. Due to the high incidence of lung cancer in the last 10 years, the interest to study various molecular mechanisms underlying this malignant disease rapidly increased. Therefore, multiple preclinical models, such as non-small lung carcinoma mouse models are publicly available [66]. Based on the histology of the malignant cells, lung cancers are categorized into two large subgroups: NSCLC and small cell lung carcinoma (SCLC). NSCLC is the most common type of lung cancer and represents 85% of all lung cancer cases. NSCLC includes three large subtypes: ADC (40%), SCC (30%), and large cell carcinoma (15%). SCLC represents the remaining 15% of the lung cancer cases [67]. Considering the high rates of epithelial carcinoma tumors arising from the respiratory system, lung cancer can be considered an exceptional model to study the main pathophysiological mechanism of SCC compared to other epithelial tumors, such as ADC.

### 2.1. Mutational Landscape of SCC Tumors

The work published by the Cancer Genome Atlas Research Network focused on a detailed analysis of the mutational landscape of SCC tumors [68,69]. This extensive analysis comprises DNA copy number alterations, mutations, mRNA expression, and promoter methylation of SCC patient biopsies. These unprecedented novel insights into this disease increased our understanding of the mutational complexity of SCC tumors. SCCs present one of the genetically most complex tumors, as they vary dramatically between patients in terms of the occurrence of tumor drivers [70]. Predominantly recurring genetic alterations are found in p53; these are more frequent compared to other tumor entities [71,72]. Copy number alterations of SOX2, PDGFRA, or FGFR1, along with deletions of tumor-suppressor CDKN2A, are frequently reported for SCC. FBXW7 is commonly mutated or lost in SCCs, such as in the lungs [73], esophagus [74], head and neck [75]. In addition, recurrent mutations in RB1, KEAP1, NFE2L2, BAI3, GRM8, MUC16, RUNX1T1, LKB1, and ERBB4 have also been identified [68]. Common oncogenic drivers found in SCCs are FGFR1, KRAS, EGFR, DDR2, PIK3CA, SOX2, and BRAF [76,77].

SCCs are therapeutically challenging tumor entities. These tumors are frequently aggressive and highly metastatic. Low response rates to radiotherapy and chemotherapy are commonly observed in SCC patients, and accordingly, the mortality rate is particularly high in lung SCC patients, and ~400,000 patients die owing to lung SCC every year. Nevertheless, considerable progress has been achieved due to the development of spiral low-dose computed tomography (LDCT) screening for early diagnosis of lung cancer, which could reduce the mortality of lung cancer patients by 20% [78].

### 2.2. Tp63 in Squamous Cell Carcinoma

Another key player in human SCC oncogenesis is the proto-oncogene Tp.63 One of the most recurrent and common genomic alterations observed in SCCs of different entities is the amplification of chromosome 3 in the region located between 3q26 and 3q28, which includes the Tp63 gene [79]. Based on its structure, Tp63 is a member of the p53 superfamily of transcription factors. Previous phylogenetic analyses showed that Tp63 is the founding member of this family [80]. The Tp63 gene is significantly conserved between species, and invertebrate organisms, such as *Caenorhabditis elegans* or *Drosophila melanogaster,* express a similar protein. p53, Tp63, and Tp73 contain N-terminal transactivation domains (TADs), DNA-binding domain (DBDs), and oligomerization domains (OD). Additionally, the C-terminus region of Tp63 contains a sterile alpha motif (SAM) and a transactivation inhibitory domain (TID) (Figure 4A). p53, Tp63, and Tp73 execute their transcriptional functions as homotetramers (four molecules of p53, Tp63, or Tp73) or heterotetramers (tetramers comprising heterogeneous combinations of p53, Tp63, or Tp73 proteins) through their ODs [81]. Tp63 exhibits a high preference for forming heterotetramers with Tp73 rather than p53. Members of the p53 superfamily can interact with and regulate the functions of each other via multiple mechanisms. For example, the DBD domains from mutant and wild-type (WT) p53 can bind to Tp63 and Tp73, thereby regulating their transcriptional activity [80,82,83].

There are two predominant variants of the Tp63 protein, generated using two alternative promoters, TAp63 and ΔNp63 (Figure 4B) [81]. The TAp63 protein contains a consensus transactivation domain at its N-terminal region. This N-terminal consensus transactivation domain located in TAp63, TAp73, and p53 regulates proapoptotic signaling [82]. In contrast, ΔNp63 variant lacks this domain, and acts as a dominant negative regulator (similar to mutant p53) toward TAp63, TAp73, and p53. Overall, 10 different isoforms arise via alternative splicing at the C-terminal region. These are p63α, β, γ, δ, and ε for TAp63 and ΔNp63 proteins [81]. p53 is expressed ubiquitously, whereas p63 and p73 are tissue-specific isoforms. The main isoform expressed in adult squamous tissues and SCC is ΔNp63α, containing a truncated p53 TAD at the N-terminal region, along with a second transactivation domain encoded mainly by exon 11 (Figure 4A,B) [81].

The protein turnover of ΔNp63α is regulated by the ubiquitin system [84]. Several E3-ligases are involved in the regulation of its stability, localization, and activity. Thus far, the E3-ligases FBXW7, ITCH1, WWP1, PIRH2, and NEDD4 are reported to ubiquitinate ΔNp63α, resulting in its proteasome degradation [81,84]. Additionally, MDM2 monoubiquitinates ΔNp63α, which impacts its transcriptional activity by affecting protein localization. Following MDM2 ubiquitination, ΔNp63α translocates from the nucleus to the cytosol [29]. The E3-ligase-dependent regulation of ΔNp63 in SCCs has clinical implications, as FBXW7 is commonly mutated or deleted in SCCs, leading to increased ΔNp63 protein abundance and stability [73]. Recently, USP28 was identified as the first DUB able to regulate ΔNp63 protein stability [17].

ΔNp63α can regulate its target gene expression by directly binding to the promoters of its target genes, resulting in “canonical” target gene activation or repression. Alternatively, ΔNp63α can interact with other transcription factors, regulating their function or affinity toward consensus binding sites [85]. Furthermore, ΔNp63α can modify chromatin accessibility upon its attachment to enhancer sites and binding to chromatin remodeling factors [86], as well as regulate the processing and formation of miRNAs [87]. ΔNp63α globally regulates gene expression and signaling networks through multiple processes and mechanisms.

Recent studies have extensively elucidated the function of ΔNp63 in SCC pathogenesis. These studies report that the increased expression of ΔNp63α regulates a high number of signaling pathways and genetic rearrangements that crucially contribute to SCC oncogenesis [82]. ΔNp63α induces a specific genetic transcriptional profile that completely transforms genetic homeostasis and molecular processes. ΔNp63α regulates key factors in tumor development, such as EGFR, MAPK, and T-cell receptor signaling pathways; chromatin remodeler pathways; and WNT, BMP, TGF-B, and NOTCH, thereby significantly contributing to oncogenic transformation and tumor onset. Additionally, ΔNp63 is involved in tumor induction and maintenance functions, such as cell adhesion, proliferation, apoptosis, DNA repair, epithelial mesenchymal transition (EMT), glutathione metabolism, and cellular redox processes or senescence [88]. Furthermore, the oncogenic potential of ΔNp63α collaborates with the RAS pathway to induce tumorigenesis [89].

Additional studies have observed that ΔNp63α is an indispensable transcription factor for the induction of oral SCC in vivo [90]. Recently, in pancreatic cancer, ΔNp63α was identified to induce the trans-differentiation of PDAC to the squamous subtype by enhancer cellular reprogramming [91]. Hence, SCCs, are addicted to ΔNp63 expression, and its depletion or loss triggers a proapoptotic program and suppresses SCC proliferation. One cannot eliminate the possibility that activation of the programmed cell death upon genetic depletion of ∆Np63 is mediated by direct interference with the activity of TAp63, TAp73, or p53 [85,92]. However, previous studies have demonstrated that the loss of p53 or TAp73 does not counteract the cell death, and reduced proliferation caused by ΔNp63 depletion in SCC cells [93,94,95].

Low response rates to traditional cancer therapies, such as chemotherapy, are particularly common in SCC patients. Although initial responses to these treatments are observed, SCCs frequently acquire additional mutations, activating alternative signaling cascades that can restore tumor cell survival and proliferation [73]. This effect is partially mediated by the SCC transcription factor ∆Np63. ∆Np63 enhances the transcription of DNA repair genes involved in recombinational repair (RR) and Fanconi anemia (FA) pathways [96,97].

### 2.3. Targeting Tp63 in Squamous Cell Carcinoma

SCC patients are currently treated with the same conventional Cisplatin therapy as they would have been treated with 30 years ago. In contrast to patients with other tumor entities, the survival of SCC patients is limited, and the efficacy of the current therapies is rather low. Considering the rising incidences of these tumor entities, the development of novel SCC therapies is urgently required.

The essential functions of ∆Np63 in SCC tumors have been extensively studied. High levels of ΔNp63 protein abundance are essential to induce and maintain SCC tumors [90,92]. ΔNp63 is a crucial protein to maintain the malignant transcriptional profile and SCC identity of epithelial tumors [17,91]. However, ΔNp63 is also critical for maintaining genome stability, and its depletion strongly increases DNA damage and apoptosis in SCC tumors [92,97]. Considering the limited portfolio of effective therapies available to treat SCC tumors, targeting ∆Np63 has emerged as potential alternative to improve the therapeutic response and clinical outcomes of SCC patients.

However, ∆Np63 is a transcription factor and as with most transcription factors, it is considered undruggable [98]. The structure of the transcription factors does not provide suitable domains for the binding of small molecule inhibitors. [99,100]. The fact that ∆Np63 directly or indirectly regulates a massive subset of different genes and cellular processes makes it almost impossible to completely block the ∆Np63 SCC profile targeting downstream effectors. DUB inhibitors that can regulate the intracellular protein degradation in specific cancer-associated targets have emerged as novel therapeutic options for cancer treatment [35,101,102]. Considering that ∆Np63 is tightly regulated by the UPS, for the first time, ∆Np63 was targeted in vivo using a small molecule inhibitor targeting the activity of USP28. The pharmacological inhibition of USP28 was sufficient to hinder the growth of SCC tumors in preclinical mouse models [17].

## 3. USP28: Oncogene or Tumor Suppressor? The Importance of Genetics

USP28 is a deubiquitylase that is extensively characterized in several tumor entities, such as colorectal cancer [21,33]. However, its function in SCC tumors was just recently elucidated [17]. In SCC tumors, the inhibition of USP28 reduces tumor growth and increases cell death [17]. Previous studies have shown that USP28 also acts as an oncoprotein in different tumor entities, such as glioblastoma [103] and breast cancer [41].

USP28 can be considered a strong oncoprotein that stabilizes the SCC transcription factor ∆Np63 and other ubiquitous potent oncogenic proteins, such as c-MYC, c-JUN, or NOTCH1 (Table 2; Figure 5A,B). In addition, USP28 stabilizes ZNF304 leading to the hypermethylation and transcriptional silencing of tumor-suppressor genes upon oncogenic transformation [20]. USP28 also regulates angiogenesis, metastasis and cell adaptation to hypoxia stabilizing HIF-1α [42]. USP28 increases aerobic glycolysis and promotes cell proliferation upon stabilization of FOXC1 [50]. Moreover, USP28 also stabilizes other oncoproteins (Table 2), such as STAT3 [48] or LIN28 [49], promoting cancer cell viability and growth.

It has been demonstrated that USP28 stabilizes p53 and other tumor-suppressor proteins acting as tumor suppressor [18,44] (Table 2). Previous reports showed that 53BP1 and USP28 activate p53, preventing the proliferation of cells that have an increased chance of mitotic errors [104]. Furthermore, Caspase-8 cleaves and inactivates USP28 to overcome the p53-dependent G2/M checkpoint in cancer cells [18]. USP28 also maintains genome stability upon the regulation of other DNA damage substrates such as MDC1. Furthermore, previous research shows that USP28 stabilizes CHK2 counteracting the E3-ligase PIRH2 [36]. Considering that CHK2 phosphorylates and stabilizes p53, inducing its apoptotic activity upon DNA damage, it can be deduced that USP28 acts as a tumor suppressor when it increases the stability of CHK2. Lastly, USP28 also acts as a tumor suppressor, inducing H2A-mediated transcriptional activation of tumor-suppressor genes [43].

In summary, USP28 stabilizes oncogenic transcription factors, such as c-MYC, C-JUN, or NOTCH1; it also regulates tumor-suppressor proteins such as p53 or CHK2 (Table 2). Hence, is USP28 a tumor suppressor or an oncogene? To answer this question, we must consider the nature of the genetic alterations occurring in cancer. In particular, the genetic status of p53 and FBXW7 are highly important to elucidate the role of USP28 in a specific cell-type.

### 3.1. USP28 and p53 Genetic Status

As indicated, recent reports have demonstrated that USP28 stabilizes and regulates the activity of p53, acting as tumor suppressor [18,44,45,105]. SCC tumors frequently suffer p53 mutations or genetic alterations [71,72]. In tumors with functional mutations in p53, the stabilization of p53 by USP28 may even prove detrimental for cancer patients, because mutant p53 is considered an oncogene. Mutant p53 acts as a negative regulator of the tumor suppressor p73 [106] and positively regulates the frequency of metastasis and therapy resistance [107]. In tumors with deleted or mutant p53, the proapoptotic function of CHK2 cannot be accomplished [108] and the stabilization of CHK2 by USP28 may not have tumor-suppression functional consequences. One cannot exclude the possibility that USP28 acting as a tumor suppressor regulates other members of the p53 family, such as Tp73 or TAp63, in p53-deficient tumors. However, to date, no studies have demonstrated that USP28 stabilizes the tumor suppressors Tp73 or Tap63.

In differentiated cells, the expression of USP28 targets involved in stemness or malignant transformation is reduced and USP28 may act as a tumor suppressor, stabilizing targets, such as p53 and CHK2 (Figure 6A). Furthermore, the regulation of DNA damage by USP28 can be a double-edged sword, depending on the cell scenario. In somatic cells, USP28 stabilizes p53, allowing programmed cell death and preventing mitotic errors; it also maintains genome stability, thus avoiding the accumulation of oncogenic mutations upon regulation of DNA damage substrates such as CHK2, 53BP1, MDC1 or CLASPIN. Therefore, it is possible that USP28 helps maintain genome stability and facilitates apoptosis via p53, causing difficulties in the malignant transformation of somatic cells. Alternatively, in already transformed cancer cells and particularly in SCC cancer cells, USP28 acts as an oncoprotein facilitating cancer homeostasis and proliferation via stabilization of oncogenic substrates, such as c-MYC, c-JUN, NOTCH1, ∆Np63, and CCNE (Table 2). However, we cannot exclude the possibility that USP28 could act as a tumor suppressor in other tumor entities with functional p53.

### 3.2. Are USP28 Substrates Recruited via Another E3-Ligase in FBXW7-Deficient Cells?

As indicated, FBXW7 is commonly mutated or lost in SCCs [73,74,75]. Additionally, the reduced expression of FBXW7 correlates with aggressive tumors and resistance to therapy [73,75]. Considering the strong functional interconnection between DUB USP28 and E3-ligase FBXW7 (Figure 2), the genetic status of FBXW7 may be important to clarify the role of USP28 as an oncoprotein or tumor suppressor.

Previous studies have proposed that USP28 substrates are recruited via FBXW7, and therefore, their interaction with USP28 is less efficient in FBXW7-deficient systems [27,34,109]. The so-called “piggyback” model suggests that FBXW7 is required for USP28 substrate recognition; consequently, USP28 would only be able to stabilize substrates in the presence of functional FBXW7 [27]. Alternatively, previous reports indicated that USP28 can stabilize FBXW7 substrates, such as c-MYC, c-JUN, or NOTCH1, in FBXW7 knockout animals [33]. It was also showed that USP28 stabilizes ∆Np63 independently of the FBXW7 CPD phospo-degron [17]. Additionally, USP28 strongly regulated ∆Np63 in a functionally inactivated FBXW7 cell line system [17] (A431 cell line; S462Y homozygous mutant). 

Considering the disparities among the results, three alternative models are proposed (Figure 7A–C). In the first model, another E3-ligase would act as FBXW7 in the “piggyback” model and the hypothetical second E3-ligase would be required for USP28 substrate recognition (Figure 7A). USP28 counteracts PIRH2 in the regulation of CHK2 [36]. Similar to FBXW7, USP28 forms a complex with PIRH2 and CHK2, thus antagonizing the ubiquitination of CHK2 by PIRH2.

The protein PIRH2 (also called RCHY1) is a promising candidate for maintaining the previously demonstrated USP28 “piggyback” model. It is possible that PIRH2 is required for USP28 substrate recognition in FBXW7-deficient systems. Therefore, USP28 would only be able to stabilize substrates in the presence of functional FBXW7 or PIRH2. Additionally, several USP28 substrates, namely p53, CHK2, c-MYC, and ∆Np63, are polyubiquitinated by PIRH2 [36,110,111,112]. Similar to USP28, the role of PIRH2 has been a subject of controversy in cancer treatment [113]. As a regulator of p53 and CHK2, PIRH2 was considered a tumor-suppressor gene [110]. However, PIRH2 knockout animals exhibited increased levels of c-MYC and 25% of the animals developed spontaneous solid tumors [112]. 

Notably, 60% of PIRH2^−/−^ and p53^−/−^ compound germline knock out mice developed tumors and lived for a significantly shorter time than PIRH2^−/−^ or p53^−/−^ single knock-out animals [112]. The above result reconfirms that the role of E3-ligases or DUBs as oncogenes or tumor-suppressor proteins depend on the cellular context and cancer genetic [114].

It is not possible to exclude the possibility that E3-ligases other than FBXW7 or PIRH2 could form a complex with USP28, allowing for USP28 substrate recognition. A potential candidate is KLHL2, USP28 interacts to UCK1 via KLHL2 counteracting the ubiquitination of the ligase [47]. Another potential candidate is UBR5, an E3 ligase that interacts with several reported USP28 substrates such as CHK2, p53, and c-MYC [115,116,117,118]. Similar to USP28, it is not clear if UBR5 functions as an oncoprotein or a tumor suppressor [119].

The second alternative model proposed is a simpler one, wherein USP28 interacts with and deubiquitinates substrates independently of E3-ligases (Figure 7B). USP28 need not form a complex with another E3-ligase to interact with the target proteins. There are a multitude of DUBs that can interact with their substrates without forming a complex with E3-ligases [3], and it is possible that USP28 can deubiquitinate substrates independently of E3-ligases. Furthermore, it is possible that USP28 activity and substrate affinity are regulated in a tissue-specific manner and, depending on the target tissue, USP28 may or may not require an interaction with another E3-ligase for substrate recognition.

It could also be possible that posttranslational modifications regulate USP28 activity, thus enhancing substrate recognition and switching from the proposed first “piggyback” model to the second one (Figure 7C). As previously indicated, it has been proved that phosphorylation of USP28 on serine 67 and serine 714 increases its enzymatic activity upon DNA damage [10]. In brief, unphosphorylated USP28 requires complex formation with another E3-ligase, such as FBXW7, for substrate recognition, whereas phosphorylated USP28 can interact and deubiquitinate substrates independently of E3-ligases (Figure 7C).

It cannot be excluded the possibility that kinases other than ATM could phosphorylate USP28 under DNA damage stress. The downstream ATM kinase CHK2 could be a potential candidate to phosphorylates USP28 following DNA damage. It was previously shown that USP28 interacts with CHK2 [36]. However, analyses of the USP28 sequence did not reveal any CHK2 phosphorylation motif, suggesting that USP28 cannot be phosphorylated by CHK2. 

The model based on posttranslational modifications reinforces the existence of the “piggyback” model in somatic cells, while, in cells with increased levels of DNA damage or replication stress, USP28 can recognize substrates independently of ligases. As indicated, previously reported results demonstrate that USP28 can stabilize targets in FBXW7- deficient oncogenic models [17,33]. One possible explanation for the previous results can be that highly proliferative malignant cells frequently present increased levels of DNA damage and replication stress. The increased basal levels of DNA damage presented in cancer cells could induce phosphorylation of USP28, thereby allowing the stabilization of USP28 substrates independently of E3-ligases in cancer cells. Alternatively, in cells with low levels of DNA damage, such as non-transformed cells, USP28 remains non-phosphorylated, and E3-ligases are required for USP28 substrate recognition. It is possible that the applicability of the above-mentioned models is tissue-specific, considering that some tissues have a higher propensity to accumulate DNA damage than others. 

Moreover, it is possible that posttranslational modifications other than phosphorylation can also regulate the substrate recognition and deubiquitinating activities of USP28. For example, SUMOylation of the N-terminal region of USP28 negatively regulates its enzymatic activity [4,14]. One cannot exclude the possibility that different posttranslational modifications to substrates could regulate USP28 target recognition. Further studies are required to clarify the mechanisms involved in USP28 substrate recognition and deubiquitinating activities. Finally, the applicability of one model or another could be also substrate-specific; some substrates are easily recognized by USP28 while others can only be recognized via E3-ligase or upon posttranslational modifications.

## 4. Inhibitors of USP28 for SCC Cancer Therapy: Progress and Perspective

USP28 is a targetable DUB enzyme that stabilizes crucial oncoproteins in cancer, such as the transcription factor c-MYC [27]. C-MYC is considered as an essential oncoprotein and it is dysregulated in most tumors [120,121]. However, it is a transcription factor and considered “undruggable” [98]. Targeting “undruggable” transcription factors by dysregulating their protein stability via DUB inhibition could be a feasible alternative. Therefore, it is not surprising that different pharmaceutical companies and research groups have tried to develop efficient USP28 inhibitors as drugs to target cancer cells [35,122,123,124,125].

The inhibition of USP28 was particularly effective in the treatment of SCC tumors using murine models [17]. Considering the reduced portfolio of SCC personalized therapies and limited survival rates, the development of strong and specific USP28 inhibitors seems a promising approach to treat human SCC tumors.

AZ1 is the first established USP28 inhibitor [123]. AZ1 is a dual USP25/USP28 inhibitor that directly interacts with the catalytic domain of both DUBs. Its efficiency has been proved by several studies that used AZ1 as a model to compare their additionally developed inhibitors [122,126]. Recent studies proved the efficiency of a USP25/USP28 inhibitor in vivo for the first time [17,127]. Another recent preprint has confirmed the safety and potential applicability of a different USP25/USP28 inhibitor in vivo [128].

As indicated, USP25 and USP28 are closely related DUBs; hence, USP28 shows a high structural homology with USP25 (Figure 1A). However, USP25 is a cytosolic protein and shows low affinity to USP28 nuclear substrates, such as c-MYC or c-JUN. Therefore, USP25 is not functionally related to USP28 and they interact with different targets. Importantly for SCC tumors, the phenotypes of ∆Np63-/USP28-depleted cells and AZ1- exposed SCC cells were highly similar, indicating that treatment with AZ1 mainly inhibits the USP28-∆Np63 axis [17]. Considering that USP25 cannot bind to ∆Np63 [17], the ∆Np63-deficient phenotype observed in SCC cells upon AZ1 exposure may have mainly resulted from USP28 inhibition occurring independently of USP25 inhibition. However, off-target effects or phenotypes resulting from the chemical structure of the inhibitor, or USP25 inhibition, may indirectly influence the expression of ∆Np63 in SCC tumors.

All USP28 inhibitors developed to date also target USP25. As recent reports described USP25 as an oncoprotein in diverse tumor entities [127,129], the co-inhibition of USP25 by the current available USP25/USP28 dual specific inhibitors for cancer therapy might, therefore, not be an issue. Nevertheless, the oncogenic function of USP25 in SCC remains unexplored and more studies are required to prove the safety of the double inhibition in this tumor entity. Accordingly, it would be interesting to develop USP28-specific inhibitors that do not affect the activity of USP25. However, their highly similar catalytic structures make this task almost impossible for traditional inhibitors. Proximity-inducing drugs (Proxidrugs) regulating intracellular degradation of therapeutic targets have emerged as a possible strategy to treat a large number of cancers that have been considered non-treatable to date [130,131]. As USP28 and USP25 only share 51% of their structures [6], it may be possible to identify and characterize suitable ligands that specifically interact with USP28, to develop a novel generation of USP28 Proxidrug inhibitors. In summary, the development of Proxidrugs, such as Proteolysis Targeting Chimeric Molecules (PROTACS) and molecular glues, which can regulate the intracellular degradation of USP28, could emerge as a promising therapeutic option for SCC tumors.

Interestingly, the two USP28 SQ/TQ motifs on serine 67 and serine 714 are not conserved in USP25. Considering the phosphorylation of USP28 on serine 67 and serine 714 regulates its enzymatic activity, it could be also possible to develop drugs targeting the phospho-sites. The USP25 activity may not be affected upon exposure to an inhibitor blocking the phosphorylation of USP28 on serine 67 and serine It could be another therapeutic option to target the binding between ATM and USP28 and reduce the phosphorylation of USP28, thus decreasing its enzymatic activity in cancer cells with increased DNA damage levels. In future, specific compounds targeting USP28 without altering the USP25 activity could be developed by using Proxidrugs or targeting specific posttranslational modifications of USP28.

## 5. Concluding Remarks 

The balance between the expression and genetic status of USP28 substrates and regulators could determine the potential role of USP28 as an oncoprotein or tumor suppressor. In SCC tumors, Usp28 can clearly be considered an oncogene because most tumors suffer from functional p53 alterations. Alternatively, SCC tumors express high levels of USP28 and the oncogenic transcription factors NOTCH1, c-JUN, c-MYC, and ∆Np63 (Figure 6B). squamous tumors depend on the USP28-∆Np63 axis to maintain SCC cell identity and induce the formation of tumors in vivo. In conclusion, SCC can be considered the perfect paradigm of a tumor entity where USP28 acts as an oncogene, but one cannot exclude the possibility that USP28 could act as a tumor suppressor in other tumor entities or cell types with functional p53 and a different genetic context. 

As previously discussed, most transcription factors are considered undruggable [98] because the structure of the transcription factors does not provide a suitable domain for the binding of traditional small-molecule inhibitors [99,100]. However, recent reports have demonstrated that is possible to target complex transcription factors in vivo using USP28 small molecule inhibitors [17,127]. DUB inhibitors have emerged as a suitable therapeutic option to drug transcription factors that have been considered undruggable to date and, therefore, the papers discussed herein serve as a proof-of-concept that it is indeed possible to regulate the intracellular degradation of potent oncogenic transcription factors, such as c-MYC, ΔNp63 or c-JUN, using DUB inhibitors.

## Figures and Tables

**Figure 1 cells-10-02652-f001:**
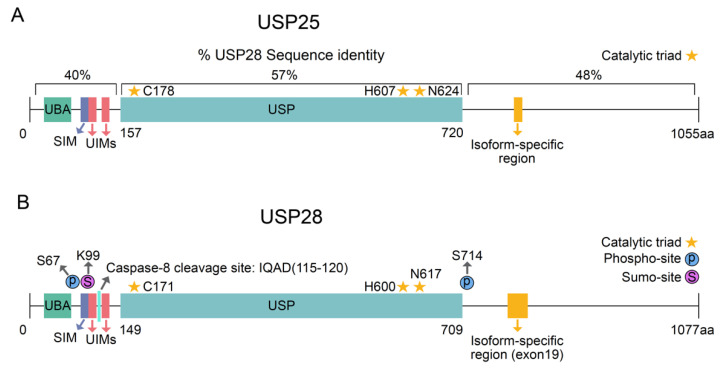
Structure of USP25 and USP28. (**A**) Schematic representation of human USP25 and USP28. The percentage of sequence identity between USP28 and USP25 for the different regions is indicated. (**B**) Schematic representation of posttranslational modifications regulating USP28 activity. UBA = ubiquitin-associated domain; SIM = sumo interacting domains; UIM = ubiquitin-interacting motif; Ubiquitin-specific peptidase domain = USP.

**Figure 2 cells-10-02652-f002:**
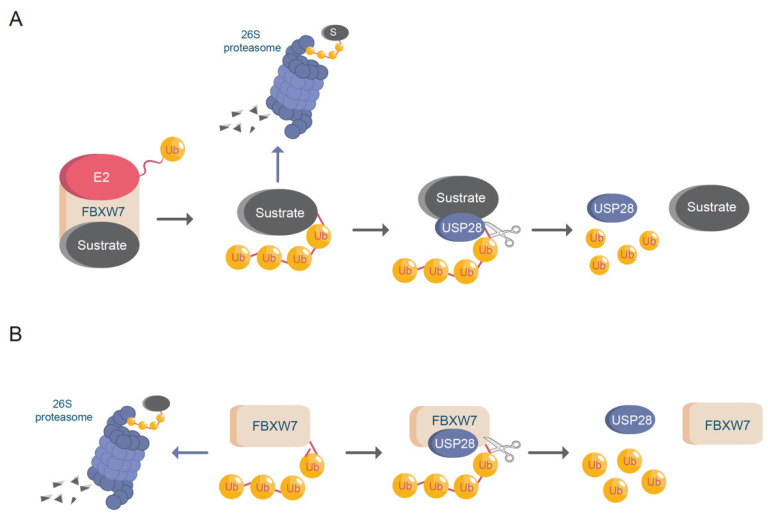
USP28 and FBXW7. (**A**) FBXW7 induces protein (substrate) ubiquitination enhancing proteasomal degradation. USP28 deubiquitinates FBXW7 substrates avoiding protein degradation. Ub = ubiquitin; E2 = ubiquitin conjugating enzyme. (**B**) USP28 increases FBXW7 protein stability via deubiquitination. Ub = ubiquitin.

**Figure 3 cells-10-02652-f003:**
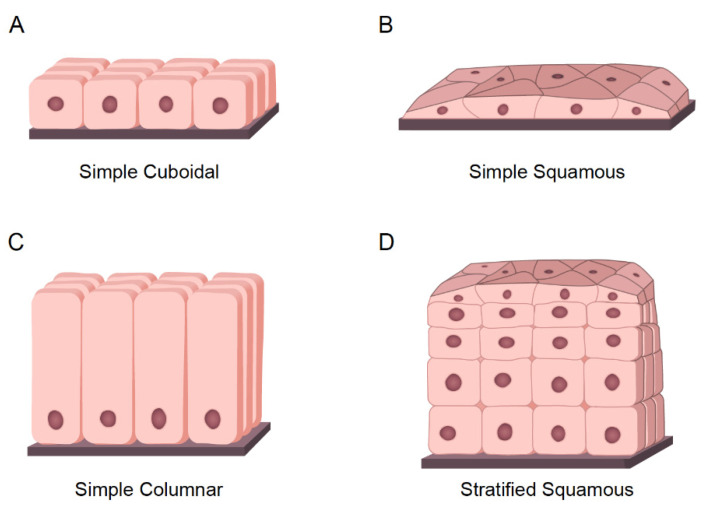
**Types of Epithelial tissues.** (**A**) Representative model of simple cuboidal epithelium. (**B**) Representative model of simple squamous epithelium. (**C**) Representative model of simple columnar epithelium. (**D**) Representative model of stratified epithelium; As an example, the model represents the stratified squamous epithelium.

**Figure 4 cells-10-02652-f004:**
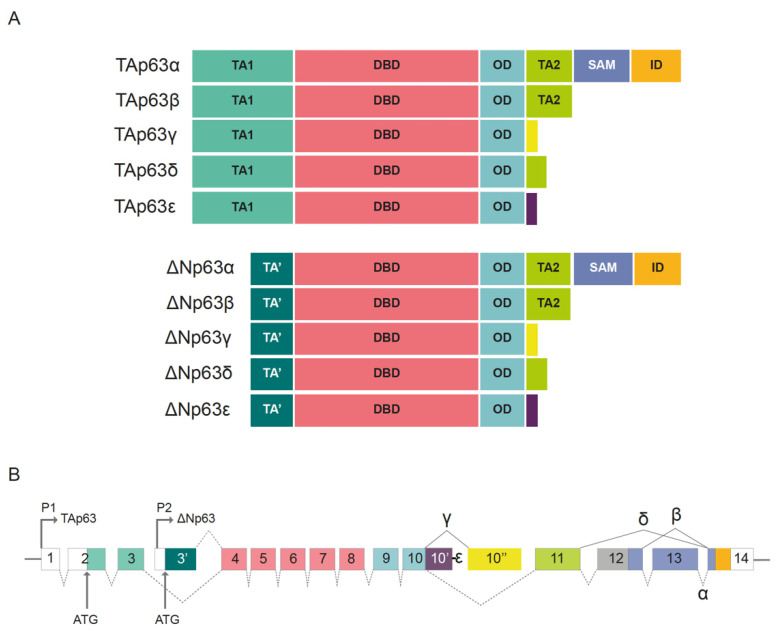
Tp63 isoforms. (**A**) TAp63 and ΔNp63 with α, β, γ, δ, ε splice variant isoforms. TA: transactivation domain; TA’: N-terminally truncated transactivation domain; DBD: DNA binding domain; OD: oligomerization domain; TA2: second transactivation domain; SAM: sterile alpha motif; ID: inhibitory domain. (**B**) Schematic representation of the P63 gene exon organization. Alternative promoters induce the synthesis of TAp63 (P1 promoter) or ΔNp63 (P2 promoter). α, β, γ, δ, ε splice variant isoforms are generated upon alternative splicing in C-terminal region. Alternatively, spliced forms for exon 10 are indicated as 10′ and 10″. Exon numbering is indicated. The exon color code corresponds to protein domains. Adapted from Vanbokhoven, Melino et al. 2011 [81].

**Figure 5 cells-10-02652-f005:**
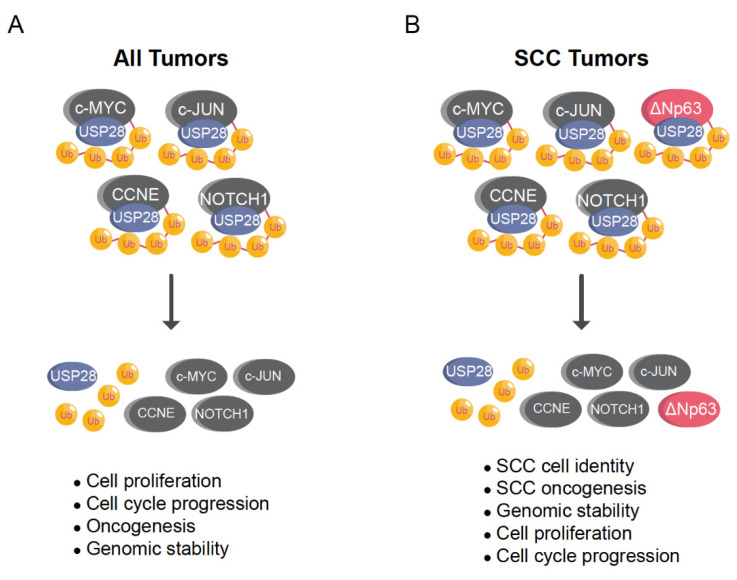
USP28 stabilizes ∆Np63 enhancing SCC cell identity, oncogenesis, proliferation, cell cycle progression and genomic stability. (**A**) Ubiquitously, USP28 regulates the stability of several oncoproteins involved in proliferation, cell cycle progression and oncogenesis, such as c-MYC, c-JUN, NOTCH1 or CCNE (**B**) In SCC tumors, USP28 regulates the oncoproteins indicated in (**A**) but also ∆Np63. ∆Np63 strongly regulates SCC cell identity, oncogenesis, proliferation, cell cycle progression and genomic stability. Ub = ubiquitin.

**Figure 6 cells-10-02652-f006:**
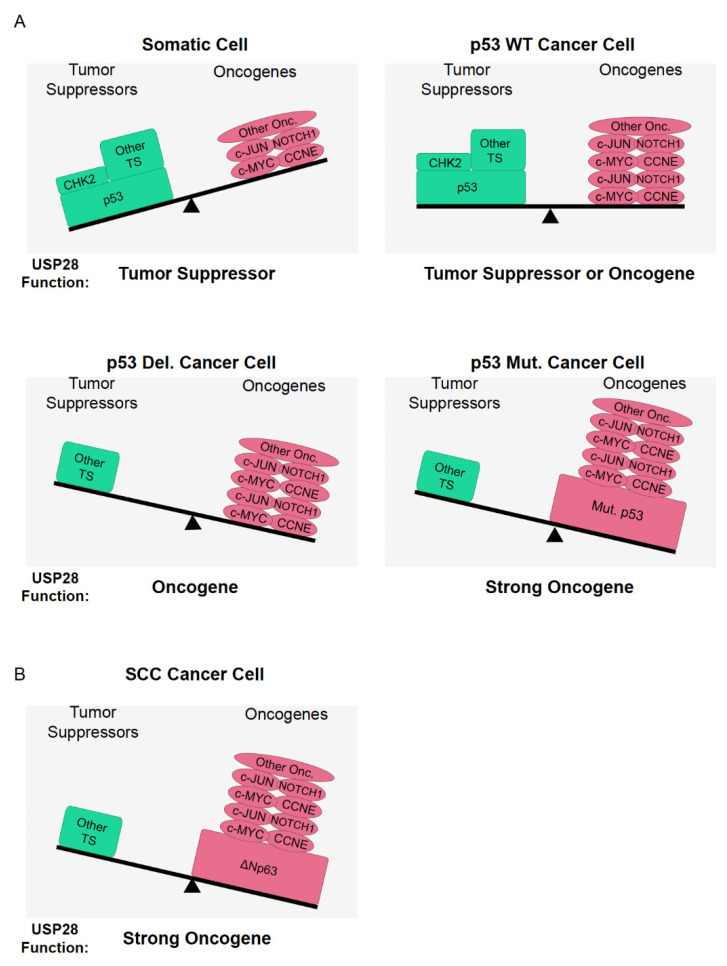
USP28 acts as an oncogene in tumors with p53 functional genetic alterations. (**A**) In somatic cells, the expression of oncogenic USP28 substrates is quite reduced and it is possible to consider USP28 a tumor-suppressor gene. In p53 wildtype (WT) cancer cells, the role of USP28 is not clear and might be determined by the expression and the genetic status of USP28 substrates. In p53 deleted (Del.) or mutant (Mut.) cancer cells, the pro-apoptotic function of CHK2 cannot be accomplished and the stabilization CHK2 by USP28 will not have functional consequences as tumor suppressor. In consequence USP28 acts as an oncogene in p53 deleted (Del.) or mutant (Mut.) cells. (**B**) In SCC cancer cells, the expression of oncogenic USP28 substrates c-MYC, c-JUN, NOTCH1 and ∆Np63 is high and p53 is frequently deleted or mutated. Accordingly, USP28 is an oncoprotein in squamous tumors. TS = Tumor suppressor; Onc = Oncogene.

**Figure 7 cells-10-02652-f007:**
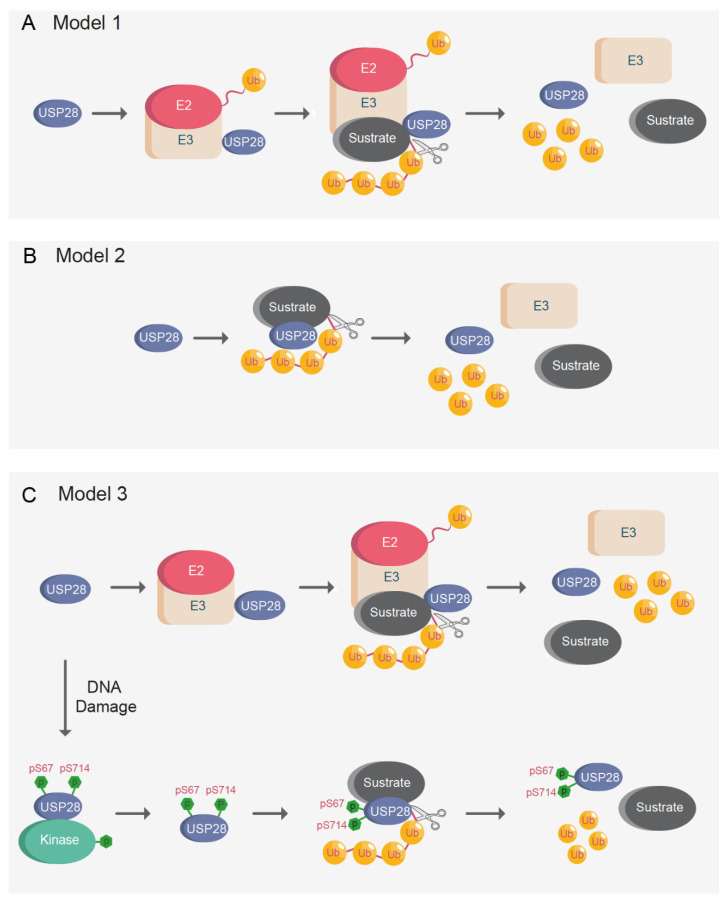
Models proposed for USP28 substrate recognition. (**A**) Model 1: An E3-ligase is required for USP28 substrate recognition. USP28 only stabilizes proteins upon the formation of a complex with the E3-ligase. (**B**) Model 2: USP28 deubiquitinates substrates independently of E3-ligases. (**C**) Model 3: Unphosphorylated USP28 acts as (**A**) (model 1). Upon DNA damage or replication stress, the phosphorylated USP28 on serine 67 and serine 714 acts as (**B**) (model 2). Ub = ubiquitin; p = phosphorylation; S = serine.

**Table 1 cells-10-02652-t001:** Regulators of USP28.

Regulator:	Positive or Negative Regulation:	Effect on USP28:	REF.:
**ATM**	Positive	Phosphorylates USP28 increasing its activity	[10]
**CASPASE 8**	Negative	Cleaves and inactivates USP28	[18]
**SENP1**	Positive	Desumoylates USP28 increasing its activity	[14]
**HDAC5**	Positive	Increases USP28 stability blocking its polyubiquitination	[19]
**miR-92b-3p**	Negative	Reduces USP28 translation	[22]
**miR-216b**	Negative	Reduces USP28 translation	[23]
**mi-R3940-5p**	Negative	Reduces USP28 translation	[25]
**miR-500a-5p**	Negative	Reduces USP28 translation	[24]
**Circ-FBXW7**	Negative	Reduces USP28 substrate recognition decreasing its activity	[26]
**c-JUN**	Positive	Increases USP28 transcription	[20]
**c-MYC**	Positive	Increases USP28 transcription	[21]

**Table 2 cells-10-02652-t002:** USP28 protein targets.

Target:	USP28 Effect:	USP28 Cancer Function:	REF.:
c-MYC	Increases its stability	Oncoprotein	[21,27,33,34,39,40]
FBXW7	Increases its stability	Tumor suppressor	[34]
c-JUN	Increases its stability	Oncoprotein	[21,33,34]
NOTCH1	Increases its stability	Oncoprotein	[21,33,34]
CLASPIN	Increases its stability	Oncoprotein or tumor suppressor dependent on cellular context	[38]
CHK2	Increases its stability	Tumor suppressor	[10,36]
53BP1	Increases its stability	Tumor suppressor	[10]
MDC1	Increases its stability	Tumor suppressor	[10]
LSD1	Increases its stability	Oncoprotein	[19,41]
HIF-1a	Increases its stability	Oncoprotein or tumor suppressor dependent on cellular context	[14,42]
CCNE	Increases its stability	Oncoprotein	[21,33,34]
H2A	Enhances transcriptional activation	Tumor suppressor	[43]
∆Np63	Increases its stability	Oncoprotein	[17]
p53	Increases its stability	Tumor suppressor	[18,44,45,46]
UCK1	Increases its stability	Tumor suppressor	[47]
STAT3	Increases its stability	Oncoprotein	[48]
ZNF304	Increases its stability	Oncoprotein	[20]
LIN28A	Increases its stability	Oncoprotein	[49]
FOXC1	Increases its stability	Oncoprotein	[50]
MTOR	Increases its stability	Oncoprotein	[34]
MCL1	Increases its stability	Oncoprotein	[34]
CD44	Increases its stability	Oncoprotein	[51,52]

## Data Availability

Not applicable.

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
