# Peer review of "USP28: Oncogene or Tumor Suppressor? A Unifying Paradigm for Squamous Cell Carcinoma"

_cells, 2021, doi:10.3390/cells10102652_

Round 1
Reviewer 1 Report
Overall a well researched and thorough body of work that will be of interest to researchers in the field. The review is concise and accessible and neatly summarizes current findings on squamous cell carcinoma and USP28. The focus on USP28 inhibitors as a potential therapeutic option for SCC is particularly relevant considering the lack of appropriate targeting therapies for this disease. Some small comments below:
- The authors highlight a role for USP28 overexpression in SCC and regulation of deltaNp63. Considering both are overexpressed in SCC have the authors noted any correlation between USP28 expression and overall patient survival or response. Possibly I have missed it or it is not available but it would be interesting to note if there was a correlation.
- Considering the authors describe in detail the role of deltaNp63 in SCC it is not readily apparent the connection to USP28. The authors do reference Prieto-Garcia paper on USP28, however it is not immediately apparent from the text the relationship between USP28 and deltaNp63. I think the authors could flesh out this a little more, considering the have a lot of text on deltaNp63.
- Do the authors know if any of the USP28 inhibitors have progressed to clinical trials? Also considering the over target effects on USP25, can the authors speculate if this will be in issue?
- There are formatting issues with the citation style thought the text. The authors should double check that the style is consistent and follows journal guidelines. In some areas it is difficult to know if the authors are discussing an amino acid or if it is a citation.
- Some language tweaks would improve flow.
Author Response
We would like to thank the reviewers for their overall friendly and very positive evaluation of our manuscript and their constructive suggestions. We have addressed the points raised by the reviewers and strongly believe that we thereby increased the quality of our manuscript. Please find the detailed replies below.
Reviewer 1.
Overall a well researched and thorough body of work that will be of interest to researchers in the field. The review is concise and accessible and neatly summarizes current findings on squamous cell carcinoma and USP28. The focus on USP28 inhibitors as a potential therapeutic option for SCC is particularly relevant considering the lack of appropriate targeting therapies for this disease.
We thank the reviewer for this very positive assessment of our work and his/her support.
Some small comments below:
The authors highlight a role for USP28 overexpression in SCC and regulation of deltaNp63. Considering both are overexpressed in SCC have the authors noted any correlation between USP28 expression and overall patient survival or response. Possibly I have missed it or it is not available but it would be interesting to note if there was a correlation.
The survival data was previously reported in a publication (Prieto-Garcia et al. 2020). As this is an important point, we included a statement regarding patient survival data in the updated version of the manuscript:
‘’USP28 gene expression correlated with poor prognosis and shortened lifespans in SCC patients 17.’’
Considering the authors describe in detail the role of deltaNp63 in SCC it is not readily apparent the connection to USP28. The authors do reference Prieto-Garcia paper on USP28, however it is not immediately apparent from the text the relationship between USP28 and deltaNp63. I think the authors could flesh out this a little more, considering the have a lot of text on deltaNp63.
We appreciate the suggestion of the reviewer to improve our manuscript. We therefore elaborated in more detail the intricate relationship between USP28 and deltaNp63 in SCC. Please find the novel sections below:
“For Squamous cell carcinoma (SCC), USP28 function was recently clarified17. In SCC, USP28 is strongly expressed and stabilizes the essential Squamous transcription factor ΔNp63….”
“Notably, SCC requires high levels of the axis USP28-ΔNp63 to maintain the malignant phenotype and its pharmacologic inhibition dramatically reduces the number of SCC tumors in lung cancer mouse models17.”
“Recently, USP28 was identified as the first DUB able to regulate ΔNp63 protein stability”
“Considering that ∆Np63 is tightly regulated by the UPS, for the first time, ∆Np63 was targeted in vivo using a small molecule inhibitor targeting the activity of USP28. The pharmacological inhibition of USP28 was sufficient to hinder the growth of SCC tumors in preclinical mouse models”
“Squamous tumors depend on the USP28-∆Np63 axis to maintain SCC cell identity and induce the formation of tumors in vivo.”
Do the authors know if any of the USP28 inhibitors have progressed to clinical trials?
We don’t have knowledge of any clinical trials with USP28 inhibitors up to now. We are aware of pharmaceutical companies actively pursuing the development of DUB specific inhibitors. According to the database clinicaltrials.gov, there are currently no clinical trials initiated. We hope that our recent work and the work published by other groups contribute to the initiation of translational studies, at least in the field of lung cancer.
Also considering the over target effects on USP25, can the authors speculate if this will be in issue?
We added some sentences speculating about the function of USP25 in SCC tumors:
“All USP28 inhibitors developed till date target USP25 as well. As recent reports de-scribed USP25 as an oncoprotein in diverse tumor entities 127,129, the co-inhibition of USP25 by current available USP25/USP28 dual specific inhibitors for cancer therapy might there-fore not be an issue. Nevertheless, the oncogenic function of USP25 in SCC remains unexplored and more studies are required in order to proof the safety of the double inhibition in this tumor entity.”
There are formatting issues with the citation style thought the text. The authors should double check that the style is consistent and follows journal guidelines. In some areas it is difficult to know if the authors are discussing an amino acid or if it is a citation. Some language tweaks would improve flow.
We apologize for the formatting issues. We addressed the formatting issues and the manuscript was reviewed by a native speaker in order to improve the flow of the manuscript.
Reviewer 2 Report
This review provides an interesting perspective on the role of USP28 in Squamous Cell Carcinoma. In particular, the authors discuss the role of USP28 in different genetic backgrounds and the use of USP28 inhibitors for the treatment of cancer even though it has a diverse set of targets, both oncogenes and tumor suppressors. Overall, the review is interesting as deubiquitinases (DUBs) are potential therapeutic targets and the authors provide insights into why the targeting of this DUB may be complex. Below are some minor points that should be addressed prior to publication.
Line 52 – the authors discuss the cleavage of Ub chains by USP28 without a proper introduction to the diversity of Ub chains. Since the statement is a “contrast to other USP enzymes”, the authors must first describe Ub chains and that DUBS have different linkage specificities. These ideas could be introduced around line 35 as the authors discuss that DUBs cleave the bond between Ub and the target protein, but do not introduce Ub chains or DUB linkage specificities.
Line 52-68 – the authors may consider reworking this paragraph. It goes through a lot of concepts without properly introducing the proteins or modifications well. This paragraph maybe challenging for novice readers. Similarly, a number of acronyms and proteins are not readily described throughout the review and could lead to confusion. For example, both APC and APC/C are discussed in the manuscript, but neither are introduced and are very different.
Given the number post-translational modifications (PTMs) that are discussed, it is my recommendation that a domain map of USP28 be added to the figures. On that domain map, the PTMs can be added and the authors can refer to the figure as they discuss the implications of PTMs on DUB function.
In the concluding remarks, it would be appreciated if the authors could speculate on whether the concepts of targeting USP28 would be broadly applicable to other DUBs. Including this type of discussion may help broaden the readership beyond people that work on USP28 or SCC.
Line 97- that figure reference should only be Figure 1B.
Several references in the text are superscript but many are not.
Author Response
Reviewer 2.
This review provides an interesting perspective on the role of USP28 in Squamous Cell Carcinoma. In particular, the authors discuss the role of USP28 in different genetic backgrounds and the use of USP28 inhibitors for the treatment of cancer even though it has a diverse set of targets, both oncogenes and tumor suppressors. Overall, the review is interesting as deubiquitinases (DUBs) are potential therapeutic targets and the authors provide insights into why the targeting of this DUB may be complex. Below are some minor points that should be addressed prior to publication.
We thank the reviewer for this very positive assessment of our work and his/her support.
Line 52 – the authors discuss the cleavage of Ub chains by USP28 without a proper introduction to the diversity of Ub chains. Since the statement is a “contrast to other USP enzymes”, the authors must first describe Ub chains and that DUBS have different linkage specificities. These ideas could be introduced around line 35 as the authors discuss that DUBs cleave the bond between Ub and the target protein, but do not introduce Ub chains or DUB linkage specificities.
As the reviewer suggested, we added a paragraph to introduce the existence of various types of ubiquitin chains and indicating that DUBs could present different linkage specificities:
“Ubiquitin contains seven different lysine residues (K6, K11, K27, K29, K33, K48, and K63) and N-terminal methionine (M1) groups that serve as secondary ubiquitination sites to induce different types of ubiquitin chains on substrates. Protein ubiquitination is a re-versible process occurring through the action of deubiquitinating enzymes (DUBs) that cleave the bond between ubiquitin and the target proteins. Notably, some DUBs display a certain degree of selectivity towards specific types of ubiq-uitin chains, while others show a broad promiscuity3.”
Line 52-68 – the authors may consider reworking this paragraph. It goes through a lot of concepts without properly introducing the proteins or modifications well. This paragraph maybe challenging for novice readers.
We appreciate this suggestion and to assist in ease of reading, we restructured this paragraph. In order to simplify the concepts, we expanded this section and split it in two paragraphs, describing more in detail the different processes involved in the regulation of USP28.
“The expression and enzymatic activity of USP28 is strongly regulated by several cel-lular processes (Table 1) in a context-specific manner. In particular, posttranslational modifications tightly regulate USP28 activity (Figure 1A). Upon DNA damage, the phos-phorylation of USP28 on serine 67 and serine 714 by the kinase ATM increases its enzy-matic activity10 . Apart from phosphorylation, Sumoylation can also regulate USP28 en-zymatic activity. Sumoylation is defined as the reversible conjugation of small ubiquitin related modifier molecules (SUMOs) on a substrate protein11,12. N-terminal sumoylation of USP28 decreases its enzymatic activity4,13. SENP1 strongly desumoylates USP28 upon hypoxia, demonstrating the importance of cellular context for USP28 activity,14. As re-ported for other DUBs15,16, one cannot exclude the possibility that USP28 deubiquitinates itself, thereby avoiding proteasomal degradation and in consequence, regulating its own stability. Supporting this hypothesis, recent publications observed reduced USP28 protein levels upon pharmacological inhibition17.
Independent of posttranslational modifications, other mechanisms have been re-ported which can regulate the activity or expression of USP28 in cells (Table 1). As an example, cleavage of USP28 by Caspase-8 inactivates the DUB and is required to overcome the p53-dependent G2/M DNA damage checkpoint18. Alternatively, Histone deacetylase 5 (HDAC5) promotes USP28 stability and positively regulates the protein abundance of the Lysine-specific histone demethylase 1A (LSD1) 19. The expression of USP28 can also be transcriptionally regulated, as the oncogenic transcription factors c-JUN and c-MYC bind to the USP28 promotor and positively regulatie its expression 20,21. Lastly, it was reported that different microRnas (miR) and circular RNAs (circRNAs) regulate the activity of USP28. The microRnas miR-92b-3p, miR-216b, miR-500a-5p, and miR-3940-5p negatively regulate USP28 expression blocking its translation22–25 while the circRNA FBXW7 reduce USP28 activity decreasing substrate recognition26.’’
Similarly, a number of acronyms and proteins are not readily described throughout the review and could lead to confusion. For example, both APC and APC/C are discussed in the manuscript, but neither are introduced and are very different.
We apologize for the confusion caused by the use of not described acronyms and proteins:
“Sumoylation is defined as the reversible conjugation of small ubiquitin related modifier molecules (SUMOs) on a substrate protein”
“Alternatively, Histone deacetylase 5 (HDAC5) promotes USP28 stability and positively regulates the protein abundance of the Lysine-specific histone demethylase 1A (LSD1)”
“Furthermore, USP28 also regulates G2/M DNA damage checkpoint by preventing CLASPIN degradation upon ubiquitination of the E3-ligase anaphase promoting com-plex/cyclosome (APC/C)”
Given the number post-translational modifications (PTMs) that are discussed, it is my recommendation that a domain map of USP28 be added to the figures. On that domain map, the PTMs can be added and the authors can refer to the figure as they discuss the implications of PTMs on DUB function.
Following the reviewers suggestion, we included a new figure 1 indicating the structure and PTMs of USP25 and USP28, respectively.
In the concluding remarks, it would be appreciated if the authors could speculate on whether the concepts of targeting USP28 would be broadly applicable to other DUBs. Including this type of discussion may help broaden the readership beyond people that work on USP28 or SCC.
We fully support the suggestion of the reviewer, as we are convinced that DUBs present a suitable target structure for cancer therapy and beyond. We included a paragraph discussing the possibility to target oncogenic transcription factors using DUB inhibitors.
“As previously discussed, most transcription factors are considered undruggable because the structure of the transcription factors does not provide suitable domains for the binding of traditional small molecule inhibitors. However, recent reports have demonstrated that is possible to target complex transcription factors in vivo using USP28 small molecule inhibitors. DUB inhibitors have emerged as a suitable therapeutic option to drug transcription factors considered undruggable to date and, therefore, the papers discussed herein serve as a proof of concept that it is indeed possible to regulate the intracellular degradation of potent oncogenic transcription factors , such as c-MYC, ΔNp63 or c-JUN, using DUB inhibitors.”
Line 97- that figure reference should only be Figure 1B.
We followed reviewer indication.
Several references in the text are superscript but many are not.
We apologize for that and we already solved the issue
FIGURE1
Figure 1. Structure of USP25 and USP28.(A) Schematic representation of human USP25. The percentage of sequence identity between USP28 and USP25 for the different regions is indicated.(B) Schematic representation of human USP28. Position of posttranslational modifications regulating USP28 activity are indicated. UBA= ubiquitin-associated domain SIM = sumo interacting domains, UIM= ubiquitin-interacting motif, Ubiquitin-specific peptidase domain = USP

Reviewer 3 Report
The manuscript is well thought out and well written, presenting very plausible mechanistic explanations for the oncogenic and tumor suppressive activities of USP28. There are some minor typographic errors that require correction and a few conceptual issues that merit consideration. With respect to corrections it is not clear why the word "Squamous" should be capitalized throughout, the word "Damage" should be capitalized on line 19, or the word "Breast" in line 320. I believe upper case names such as NOTCH1 etc. by convention refer to proteins (not genes), and would be more accurately described as oncoproteins, not oncogenes (the oncogenes would be Notch1, c-Myc etc.). There are many examples of this in the manuscript. There are places in which the reference numbers are not given in superscript (lines 51, 84, 112, 144) and this can generate confusion (for example "53BP18" in line 107). In Table 1 CASPASE 8 is reported to have a positive regulatory effect, which I believe is an error.
With regard to the conceptual issues the similarity of USP28 to USP25 is mentioned in several places, and there is discussion of the conservation of the catalytic domains (an obstacle to the development of specific inhibitors) and the functional divergence attributed to alterations at other sites. There is an evolutionary explanation for this that would be worth mentioning: USP28 and USP25 are ohnologs (generated from ancient whole genome duplication as reported in Vlasschaert et al. PMID: 28177072) that have shown limited divergence following the duplication event. In this context their properties are understandable, and similar to other DUB ohnolog pairs. The current manuscript alludes to the importance of context with respect to the ΔNp63 protein, the DNA damage of USP28 regulation, and so forth, and it seems inescapable that the signalling milieu under the conditions the cell is experiencing would provide the mechanistic basis for context-specific effects of USP28. The importance of context on the oncogenic versus tumor suppressive activities of deubiquitinating enzymes was the subject of a recent review (PMID: 33415735) that seems directly relevant to the current manuscript (particularly with regard to EMT, Notch crosstalk, etc.) but was not cited.
Author Response
Reviewer 3.
The manuscript is well thought out and well written, presenting very plausible mechanistic explanations for the oncogenic and tumor suppressive activities of USP28. There are some minor typographic errors that require correction and a few conceptual issues that merit consideration.
We thank the reviewer for this very positive assessment of our work.
With respect to corrections it is not clear why the word "Squamous" should be capitalized throughout, the word "Damage" should be capitalized on line 19, or the word "Breast" in line 320.
We apologize for these formatting issues. The errors were corrected and now the capitalized words were eliminated.
I believe upper case names such as NOTCH1 etc. by convention refer to proteins (not genes), and would be more accurately described as oncoproteins, not oncogenes (the oncogenes would be Notch1, c-Myc etc.). There are many examples of this in the manuscript.
We are sorry for the errors regarding naming of proteins/genes in accordance to convention. We rectified this issue. For example, human genes are indicated with italic capitals (for example NOTCH1) and proteins are capitalized (for example USP28).
There are places in which the reference numbers are not given in superscript (lines 51, 84, 112, 144) and this can generate confusion (for example "53BP18" in line 107).
We apologize for the confusion. The issue is already solved.
In Table 1 CASPASE 8 is reported to have a positive regulatory effect, which I believe is an error.
We thank the reviewer to point out this mistake. It was corrected.
With regard to the conceptual issues the similarity of USP28 to USP25 is mentioned in several places, and there is discussion of the conservation of the catalytic domains (an obstacle to the development of specific inhibitors) and the functional divergence attributed to alterations at other sites. There is an evolutionary explanation for this that would be worth mentioning: USP28 and USP25 are ohnologs (generated from ancient whole genome duplication as reported in Vlasschaert et al. PMID: 28177072) that have shown limited divergence following the duplication event. In this context their properties are understandable, and similar to other DUB ohnolog pairs.
We appreciate the reviewers suggestion. We introduced a dedicated section to the manuscript.
“USP28 was identified through homology search for USP25, as they share 51% sequence identity. Similar to other DUB ohnolog pairs, USP28 and USP25 were generated from ancient whole genome duplication and they have shown limited divergence upon the duplication event (Vlasschaert et al. PMID: 28177072). The deubiquitinases USP25 and USP28 have similar topological structures, comprising USP domains, ubiquitin-associated domains (UBAs), sumo in-teracting domains (SIMs) and two ubiquitin-interacting motifs (UIMs) at the N-terminal region3 (Figure 1A and 1B)”
The current manuscript alludes to the importance of context with respect to the ΔNp63 protein, the DNA damage of USP28 regulation, and so forth, and it seems inescapable that the signalling milieu under the conditions the cell is experiencing would provide the mechanistic basis for context-specific effects of USP28. The importance of context on the oncogenic versus tumor suppressive activities of deubiquitinating enzymes was the subject of a recent review (PMID: 33415735) that seems directly relevant to the current manuscript (particularly with regard to EMT, Notch crosstalk, etc.) but was not cited.
We were not aware of this interesting recent paper and included it in our review manuscript. Thank you.